# In Vitro Confirmation of Siramesine as a Novel Antifungal Agent with In Silico Lead Proposals of Structurally Related Antifungals

**DOI:** 10.3390/molecules26123504

**Published:** 2021-06-08

**Authors:** Josipa Vlainić, Ozren Jović, Ivan Kosalec, Oliver Vugrek, Rozelindra Čož-Rakovac, Tomislav Šmuc

**Affiliations:** 1Ruđer Bošković Institute, Bijenička cesta 54, 10000 Zagreb, Croatia; Josipa.Vlainic@irb.hr (J.V.); Oliver.Vugrek@irb.hr (O.V.); Rozelindra.Coz-Rakovac@irb.hr (R.Č.-R.); Tomislav.Smuc@irb.hr (T.Š.); 2Faculty of Pharmacy and Biochemistry, University of Zagreb, Schrottova 39, 10000 Zagreb, Croatia; ikosalec@pharma.hr

**Keywords:** siramesine, antifungal activity, *Candida albicans*, in vitro cell experiments, ergosterol, molecular docking, QSAR, Erg2, pKi prediction

## Abstract

The limited number of medicinal products available to treat of fungal infections makes control of fungal pathogens problematic, especially since the number of fungal resistance incidents increases. Given the high costs and slow development of new antifungal treatment options, repurposing of already known compounds is one of the proposed strategies. The objective of this study was to perform in vitro experimental tests of already identified lead compounds in our previous in silico drug repurposing study, which had been conducted on the known Drugbank database using a seven-step procedure which includes machine learning and molecular docking. This study identifies siramesine as a novel antifungal agent. This novel indication was confirmed through in vitro testing using several yeast species and one mold. The results showed susceptibility of *Candida* species to siramesine with MIC at concentration 12.5 µg/mL, whereas other candidates had no antifungal activity. Siramesine was also effective against in vitro biofilm formation and already formed biofilm was reduced following 24 h treatment with a MBEC range of 50–62.5 µg/mL. Siramesine is involved in modulation of ergosterol biosynthesis in vitro, which indicates it is a potential target for its antifungal activity. This implicates the possibility of siramesine repurposing, especially since there are already published data about nontoxicity. Following our in vitro results, we provide additional in depth in silico analysis of siramesine and compounds structurally similar to siramesine, providing an extended lead set for further preclinical and clinical investigation, which is needed to clearly define molecular targets and to elucidate its in vivo effectiveness as well.

## 1. Introduction

Fungal diseases affect more than one billion people and cause approximately 1.6 million deaths per year worldwide. Beyond human infections there is also an emerging field of animal, as well as plant, fungal diseases. Moreover, climate change and global warming increase the incidence of different fungal diseases [1]. Current treatment options for fungal diseases are limited and there is a need to develop new treatment strategies and new antifungal drugs that will identify various molecular targets and employ different mechanisms of action [2]. Since the early 2000s there has been no new effective and safe antifungal drug introduced to the market, and some protocols in use are based on drugs available for more than 50 years [3]. Moreover, the resistance of *Candida* spp. to the antimicrobial class of drugs is increasing, especially in clinical settings. Some of the fungal species are extremely resistant to all three classes of antifungal drugs, including *Candida auris* [4], and infections caused by multiple resistant fungal species are very difficult to eradicate [5].

As a development of a new drug is a long and complicated process with high costs, alternative options for introduction of novel drugs into therapy include evaluation of natural products and their standardization, repurposing of old substances with known indication and with proven efficacy against similar targets, etc. Strategies are in place to address antifungal resistance and include the ongoing search for new antifungal drugs. One of the most promising areas for new molecules with antifungal activity is the pool of already known molecules that often have applications in noninfectious diseases. This approach, often called repurposing [6], could promote preclinical research and clinical trials. The latter was the aim of our present study. Based on our previous study [7], we carried out drug repurposing of the Drugbank database in search of novel antifungals. Since there are many antifungals with varying modes of action, the focus was on inhibitors of sterol biosynthesis in membranes. Machine learning and molecular docking simulations were conducted on two major protein targets in fungi. We have found several in silico lead compounds based on the seven-step filtering procedure, starting by machine learning and ending with selecting molecular docking top-scored compounds. In this study, we carry out in vitro and in silico analysis of some of these top-scored antifungal lead compounds. The aim is to (a) confirm our in silico findings and (b) to suggest an extended set of potential candidate compounds from any experimentally confirmed in vitro lead compound by in silico derivative optimization.

Compounds selected in our previous study and available for in vitro testing were talarozole, L778123, siramesine, ozagrel and MBX-2982. We have screened those candidates using several yeasts and one mold, to test these potential drugs. The *Candida* sp. is one of the predominant causes of fungal diseases in humans, also associated with nosocomial infections, and the search for novel treatment options is ongoing [8]. In addition, the biofilms formed by yeasts on abiotic surfaces and mucosal membranes are a crucial step in the self-defense of fungal cells [3]. The latter represent part of *Candida* species’ pathogenicity [9], and it should be considered when introducing novel antifungal formulations into the clinic.

In our subsequent in silico study, 3D-QSAR modeling coupled with docking was used for lead compound optimization to find a promising set of siramesine-like compounds. Siramesine (Lu-28-179; 1′-[4-[1-(4-fluorphenyl)-1H-indol-3-yl]butan-1-yl]- spiro[isobenzofuran-1(3H),4′-piperidine]), a piperidine analog that binds to σ-2 receptors (agonist) with a nanomolar equilibrium dissociation constant (Kd = 1.1 mM in humans and 2.2 nM in rats) [10], was originally developed for the treatment of anxiety and depression [11]. Although it was well tolerated and nontoxic in humans, it did not become a clinically relevant drug since its efficacy was not satisfactory to treat both diseases [11]. Molecular docking and QSAR study were conducted on both the sigma-1 receptor protein and Erg2 protein for siramesine and compounds structurally similar to siramesine in the PubChem database [12]. We carried out two independent studies with two molecular docking programs. Only compounds with in silico predicted pKi values higher than siramesine for both proteins and both programs were to be claimed as candidate compounds. Our study revealed several such potential drug candidates.

To our best knowledge, this is the first study showing the antifungal potential of siramesine and siramesine-related compounds, where siramesine itself showed remarkable potential against various clinically important *Candida* species. Our study tested the ability of siramesine to inhibit ergosterol biosynthesis and we suggest this as a possible mechanism of its action against *Candida* spp. Further studies are needed to address this in depth.

## 2. Results and Discussion

### 2.1. Antifungal Activity of Siramesine on Planktonic and Biofilm-Formed Cells

Using serial microdilution broth assay, the activity of siramesine was revealed, as shown by docking studies. As shown in Table 1, MICs for fungal species used in this study for siramesine and derived compounds were between 12.5 and 250 µg/mL while *I. orientalis* and *A. brasilliensis* proved to be unsusceptible in concentration of tested compounds up to 250 ug/mL. Compared to talarozole, ozagrel, L-778123 and MBX2982, siramesine showed the most potent antifungal activity with a MIC of 12.5 ug/mL against all planktonic *Candida* species tested in broth microdilution assay. Since the results of MIC determination involved planktonic cells, we also tested the biofilm-eradication capability of tested compounds. As seen in Table 1, MBECs ranged from 50 ug/mL to >500 ug/mL following the same pattern as with susceptibility testing using planktonic cells. Again, siramesine was found to be more potent that other test siramesine-related compounds with MBECs between 50 ug/mL and 62.5 ug/mL, depending of the *Candida* species tested.

Our results also show that siramesine has considerable in vitro antifungal activity using assays on planktonic and biofilm-formed cells, in comparison with other siramesine-related compounds. Results of MBECs values against biofilm-formed cell showed that antibiofilm concentrations were 4–5 times higher than MICs.

### 2.2. Ergosterol Content

As docking study revealed siramesine may act as an antifungal with effect on ergosterol synthesis we searched for a characteristic curve indicative of ergosterol and late sterol intermediate 24(28)-dehydroergosterol (DHE), and found a loss of physiological ergosterol biosynthesis in Candida cells. In addition, the dose-dependent decrease in the height of the absorbance peaks may be seen in samples treated with siramesine and those correspond to a decrease in ergosterol content. Figure 1 shows a decrease of ergosterol level in membranes of *C. albicans* that was induced by siramesine, dose-dependently. The lowest concentration of 0.5 × MIC reduced ergosterol nonsignificantly (reduction = 15.48%; *p* > 0.05) compared to intact (untreated) cells. However, with siramesine at concentrations of 1 × MIC and 5 × MIC, a significant drop of ergosterol biosynthesis was noted with reductions of 43.51% (*p* < 0.05) and 83.68% (*p* < 0.001), respectively. Results showed that siramesine at 1 × MIC and 5 × MIC is involved in biosynthesis of ergosterol during in vitro growth, and thus this modulation of ergost erol biosynthesis, which is crucial for homeostasis of yeast cells, indicates a possible target for antifungal activity.

Previous studies in cancer cell lines showed that siramesine, as σ-2 receptor agonist, induces growth arrest and cell death, but the exact mechanism by which this is accomplished and how siramesine and other σ-2 ligands induce cytotoxicity remains unknown. The situation is the same with the possible action of siramesine. Here we show that siramesine, but not siramesine-related compounds, may induce cell death of fungal cells, having a detrimental effect on modulation of biosynthesis of ergosterol as one of the major components of the yeast membrane. Ergosterol is involved in the arrangement of lipids in a fungal membrane and it maintains membrane rigidity and fluidity, prevents penetration of water and is a crucial component that maintains membrane integrity [13,14]. The majority of sterol is synthesized in the endoplasmic reticulum (ER) and in lipid droplets of a yeast [15]. Reduced ergosterol levels also may alter activity of a variety of enzymes located in a membrane [16]. Some of the currently available antifungal medicinal products interact with ergosterol biosynthesis, such as terbinafine (this inhibits the enzyme squalene epoxidase encoded by Erg1). However, there are also parts of the ergosterol biosynthesis pathways inhibited by morpholine compounds (such as fenpropimorph, a fungicide used in agriculture), and include inhibited sterol C_8_-C_7_ isomerase, encoded by Erg2, and sterol reductase, encoded by Erg24 gene [17].

As siramesine may reduce ergosterol synthesis, it affects the cell viability at several levels, as ergosterol is involved not only in membranes but affects lipid rafts, vacuoles and mitochondria of Candida species. However, it has been shown that Erg2, a fungal gene encoding a C_8_-C_7_ sterol isomerase, and the sigma receptor share a 30.3% sequence identity (approx. 3%) and similarity (circa 66%) [18]. To hypothesize, every molecule with activity against C_8_-C_7_ isomerase encoded by Erg2 has a great potential to interact with biosynthesis of ergosterol pathways with consequent results of fungal cell death. 

Siramesine contains a piperidine group, which is one of the chemical groups listed as the G2 mode of action of agricultural fungicides together with morpholine and spiroketal-amine [19]. It is already known that piperidine compounds are potential inhibitors of the sigma-1 receptor [20] and Erg2 (delta-8 to delta-7 isomerase) protein [18], because of pharmacological similarity to the sigma-1 receptor and with mutually correlated Ki values [21], and with the amino acid sequences being 30% identical. The experimentally determined pKi of siramesine bound to the sigma-1 receptor equals pKi = 8 [22]; the sigma-1 receptor is structurally very similar to Erg2, although siramesine preferably inhibits the sigma-2 receptor. In homology modeling of Erg2 of the *Naegleria fowleri* pathogen, the 5HK1 protein was already in use as a template [23]. Unlike piperidine compounds, morpholine compounds could be both inhibitors of Erg2 and Erg24, e.g., fenpropimorph [17,24]. Other targets, such as Erg11 (C14-demethylase) are inhibited by an inhibitor with an imidazole or triazole containing group (e.g., fluconazole), while Erg1 (squalene-epoxidase in sterol biosynthesis) inhibitors (such as terbinafine) contain allylamine or thiocarbamate groups, and Erg27 (3-keto reductase) inhibitors are hydroxyanilides or amino-pyrazolinones [19]. Azole derivatives show great potential activities against *Candida* infections by targeting Erg11. Hamdy et al. developed a series of novel S-alkylated azole derivatives (compounds UoST5, 7, 8 and 11) that showed significant antifungal activities against different types of *Candida* spp. [25].

For other targets mentioned in the ergosterol biosynthesis chain [17,23] such as Erg7, Erg9, Erg25, Erg26, Erg6, Erg3, Erg4 and Erg5, there is no record in literature yet that any of the mentioned targets is inhibited by a piperidine group-containing inhibitor. Therefore, since siramesine is an antifungal that blocks ergosterol biosynthesis (which we proved), we highly suspect that siramesine’s target is Erg2.

Due to emerging resistance of fungal species to known drugs, a viable option is the use of essential oils. For example, *Ruta graveolens* essential oils caused irreversible cellular membrane damage, inducing leakage of the intracellular compounds and biofilm eradication [26]. Authors also showed synergistic action of this oil with amphotericin B. In addition, studies showed good antibiofilm properties of lavandula essential oil [27] and berberine [28] as a possible pool of novel drugs to treat candida-related infections.

It seems unlikely that the molecular mechanisms of siramesine’s effect as an antifungal compound are linked solely to the sigma receptors, and the reduction of ergosterol biosynthesis, since *C. albicans* can survive the inhibition of ergosterol synthesis, at least to some level. Another antidepressant, sertraline, showed effects on cellular physiology, including, most curiously, cytotoxicity in yeast [29,30], by an unknown mechanism that does not involve hSERT, suggesting the existence of one or more heretofore unrecognized secondary drug targets that may be evolutionarily conserved [31,32], and thus further studies are needed to elucidate this.

There are three classes of antifungal therapeutics available in clinics for yeast infection treatment (azoles, polyenes and echinocandins) and the need for novel treatment options due to emerging resistance of fungal species is growing. The high costs and research and development durability of new drug development raise awareness of drug repositioning, reprofiling and retasking. The idea of repurposing of medicinal products with marketing authorisation for an indication which does not include antifungal activity is well known. A recent review of this topic [33] showed that more than 740 articles on “antifungal drug repurposing” and similar keywords were published to October 2020, and includes both in silico/computational and experimental in vitro/in vivo approaches. In a review article, Kim et al. [33] found that more than 20 commercially available medicinal products have antifungal activity against medically important *Candida* species that includes inhibition of planktonic growth, filamentation, biofilm formation and even synergistic in vitro activity with clinically relevant antimycotics.

### 2.3. Docking and QSAR Modeling of Siramesine-Like Compounds

#### 2.3.1. Docking to Sigma-1 Receptor

Results for validated prediction of compounds’ pKi values on the test Sig-W set are displayed in Table 2 (17 test samples), while Appendix A denotes cross-validated results on the training set (66 training samples). The predictions of compounds’ pKi values are promising. The overall model performances of uninformative-variable elimination partial-least square regression (UVE-PLS) on Sig-W for both Gold and Autodock are presented in Table 3, where it can be seen that the obtained *R*^2^(val) was higher than corresponding results in [20]. This is because in this study, 1D and 2D descriptors were also considered, while in [20] the prediction model was based only on 3D descriptors. Table 4 represents the predicted pKi of a siramesine extrapolation set (S61-EX) when bound to the sigma-1 receptor protein. The experimentally determined pKi of siramesine equals pKi = 8 [22]. Since experimentally determined Ki values also vary (e.g., haloperidol 1–40 nM) [22], our result for siramesine (pKi 7.24) is within reasonable expectations. Additionally, when the standard error of the model is considered (determined RMSEP of 0.44), this means that our result is within two standard errors of experimental results (pKi(pred) = 7.24, pKi(exp) = 8, pKi(exp) > pKi(model), pKi(exp) < pKi(model) + 2xRMSEP). All compounds from S61-EX with a predicted pKi higher than siramesine could be potential candidates, however, due to the model error, we selected only those compounds for a further docking cycle of the Erg2 target with a predicted pKi higher than the prediction for siramesine, plus one standard error of the model. All such compounds are marked appropriately in the table, and represent the Erg-S set.

#### 2.3.2. Docking to Erg2 Target

Results of the most interacting amino acid residues (see Section 3) show two of the most important residues, Tyr-105 (the most interacting residue) and Glu-174 (key H-bond residue) (two residues mutually distant 69 residues) coincide in literature regarding Erg2 in different species of *Naegleria fowleri*, where they are Tyr-163 and Glu-232 (also mutually distant 69 residues) [23]. Coincidence of distance of 69 residues is also present in the sigma-1 receptor (where Tyr-103 is one of the most interacting residues and Glu-172 a key H-bond residue) [20]. Table 5 denotes cross-validated results of UVE-PLS on descriptors of the Erg-W set taken when docked to the Erg2 protein. The higher RMSECV for the Erg-W set (0.70) than that obtained for the Sig-W set (0.44) (Table 3) is likely because of the following reasons: the Erg-W set contained 44 compounds, while Sig-W had 83; the references used for Erg-W were not the most recent so the experimental error for some pKi values of these 44 compounds was significant; also for some compounds in Table 5, the pKi was approximated to be as stated although in literature it is lower [18,21]. In the end, the attained cross-validation error (Adt (0.70), Gold (0.83)) (Table 3) was still comparable with the error obtained in literature (0.50–0.81 [20], 0.72 [34]), and the cross-validated correlation coefficient (*R*^2^(CVtr)) was higher than stated in cited references due to the higher pKi range used in this study. From Table 5, it can be seen that except in a few instances (buflumedil, BM-15766) the predicted pKi values are roughly acceptable.

Attained models were used to predict the Erg-S set, and Table 6 represents the results of such prediction. Marked compounds are those with a predicted pKi higher than the prediction for siramesine plus one standard error of the model. So these are siramesine-related compounds with significantly different pKi values predicted from both programs and for both targets, and therefore they can be claimed as in silico candidate compounds. These compounds are (PubChem CID-s are given in brackets): S01 (9889994), S05 (10049018), S10 (10094903), S12 (10140458), S29 (10473699), S30 (10477563) and S41 (20290917). Figure 2 displays the structures of these seven candidate compounds and siramesine.

The analysis of molecular descriptors reveals the following information regarding the better-performing model (Appendix A): among 162 selected descriptors, 49 are 3D descriptors. The 10 most weighted descriptors (i.e., the highest absolute value regression coefficients): E1m, GATS3e, GATS3c, MATS6c, VR3_Dt, RDF45m, maxHBd, AATS5i, TDB9i, RDF75s.

## 3. Materials and Methods

### 3.1. In Vitro Experiments

#### 3.1.1. Determination of Antifungal Susceptibility

Minimum inhibitory concentrations (MIC) of selected compounds were assessed according to EUCAST protocol recommendations [35]. We employed several yeasts: *Candida albicans* ATCC 10231, *Candida albicans* ATCC 90028, *Candida tropicalis* ATCC 750, *Candida kefyr* ATCC 2512, *Candida parapsylosis* ATCC 22019, *Candida krusei* ATCC 14243, *Issatchenkia orientalis* ATCC 6258 and mold species *Aspergillus brasiliensis* ATCC 16404. Serial broth microdilutions of compounds (from 250 μg/mL to 0.1 μg/mL) in RPMI 1640 broth (Sigma-Aldrich, St. Louis, MO 63178, USA) with 2% *w*/*v* glucose were set into a sterile flat-bottom 96-well microtiter plate and cell suspension (5 × 10^6^) was added. Plates were incubated aerobically in the dark (48 h, 37 °C). Control wells contained 100 μL of fungal cell suspension in RMPI 1640 with 2% *w*/*v* glucose broth. Since the dilution performed according to the EUCAST recommendation formed turbidity, hence was unappropriated for spectrometric analysis, the replant technique was performed. Following incubation, a 10 μL of well content was seeded on the surface of a Sabouraud agar plate with calibrated microbiological loop. After 48 h incubation at 37 °C, MIC was defined as no visible count of colonies compared to control. The test was performed in triplicate on two different occasions and results are presented as mean ± SD.

#### 3.1.2. Minimum Biofilm Eradication Assay

The susceptibility of different yeast strains to siramesine and siramesine-derived compounds was tested by determining the minimum biofilm eradication concentration (MBEC). Each well (96-well plate), pretreated with FBS (fetal bovine serum, Sigma Aldrich, MO 63178, USA) (250 µL per well) was filled with 100 µL of yeast cell suspension (5 × 10^6^ CFUs/mL). Serial broth micro-dilutions of compounds (from 125 μg/mL to 0.1 μg/mL) in RPMI 1640 broth (Sigma-Aldrich, MO 63178, USA) with 2% *w*/*v* glucose were set into a sterile flat-bottom 96-well microtiter plate and cell suspension (5 × 10^6^) was added. Negative controls contained broth only. The plates were covered and incubated aerobically for 48 h at 37 °C. Following the incubation period, each well was aspirated, washed three times with PBS and vigorously shaken to remove all nonadherent cells. The remaining attached cells were fixed (100 µL methanol per well, for 15 min) and plates left to dry overnight. Formed biofilm was stained using freshly prepared crystal violet (1% *w*/*v*, 5 min). Excess stain was rinsed by placing the plate under running tap water and plates were left to dry. Adherent stained cells were solubilized (150 µL ethanol per well). The absorbance was read at 570 nm (Infinite 200, Tecan, 8708 Männedorf, Switzerland). Tests were performed in triplicate on two different occasions and results are presented as mean ± SD. The MBEC value represents the lowest dilution of a compound at which yeast/mold fails to grow.

#### 3.1.3. Modulation of Ergosterol Biosynthesis

Inhibition of ergosterol synthesis by siramesine was determined in a mass of *C. albicans* cells during the exponential growth phase. Since other compounds showed effect on yeast species at relatively high concentrations (see Table 1), we conducted these experiments using *Candida albicans* ATCC 90,028 cells as a model for in vitro ergosterol modulation. In experiments we used different concentrations of siramesine (0.5 × MIC, 1 × MIC and 5 × MIC). Positive control for the study was a sample treated with voriconazole (4 μg/mL, Pfizer, New York, NY 10017, USA). Yeast cultures were incubated at 37 °C for 18 h on orbital shaker (170 rpm) aerobically and the cells were collected by centrifugation (2700× *g*, 5 min). The pellets were weighted. Freshly prepared alcoholic potassium hydroxide solution (25% m/v, 3 mL) was added to each pellet and vortexed. Cell suspensions were transferred to borosilicate glass tubes and incubated for one hour at 85 °C in a water bath. The sterol extraction was enabled by addition of water: n-heptane mixture (1:3 *v*/*v*) followed by vortexing (3 min). Obtained heptane layer was transferred to a new borosilicate glass tube, and 0.6 mL of sterol extract was transferred to the new tube and diluted (1:5) with ethanol (100%). Samples were scanned between 240 and 300 nm at 5 nm) concentration intervals (Varian UV–VIS spectrophotometer, Agilent, Santa Clara, CA 95051, USA). In our experiments we calculated the ergosterol content as a percentage using equations:
(1)$$\begin{array}{l} \%\mathrm{ergosterol}\;+\;\%24(28)\mathrm{DHE}\;=\;A_{281.5/290}\;\times \;F)/\mathrm{cell}\mathrm{mass}\\ \%24(28)\;\mathrm{DHE}\;=\;(A_{230/518}\;\times \;F)/\mathrm{cell}\mathrm{mass}\\ \%\mathrm{ergosterol}\;=\;(\%\mathrm{ergosterol}\;+\;\%24(28)\mathrm{DHE})\;-\;\%24(28)\;\mathrm{DHE}; \end{array}$$
where F is the factor of sample dilution in ethanol (1:5) and 290 and 518 are the E values (in percentages per centimeter) determined for crystalline ergosterol and 24 (28) DHE, respectively [28].

#### 3.1.4. Statistical Analysis

The experiments were performed as triplicates at least three times on independent occasions. Results are presented as the mean ± standard deviation, where appropriate. The results of modulation of ergosterol biosynthesis were analyzed using one-way ANOVA and Tukey’s multiple comparison post-test. Statistical analyses were performed using GraphPad Prism 8.1.

### 3.2. In Silico Study

#### 3.2.1. Homology Modeling

The Erg2 protein was homologically modeled using the FASTA format from C-8 sterol isomerase protein, the gene ERG2, the organism *Saccharomyces cerevisiae* (strain ATCC 204508/S288c) (Baker’s yeast) and using the 5HK1 template. Only chain A was taken from both the obtained Erg2 protein and together with 5HK1 chain A, both chains were aligned using the T-Coffee web server for the multiple sequence alignment of protein [36]. The obtained alignment was satisfactory. After that, the structure of the obtained 5HK1 chain A was energy minimized in the Gromacs program (as described in Appendix A). Finally, the Erg2 minimized protein chain A was stripped of water and ions and 3D aligned with the corresponding 5HK1 protein chain, which can be viewed in Figure 3. What is important to emphasize is the excellent alignment of the key 5HK1 Glu172 amino acid residue with the analog Glu174 residue of Erg2, with all amino acid atom RMSD of only 1.05 Å (including hydrogens) and with alignment carboxylate oxygen OE1 position within 0.5 Å.

#### 3.2.2. Molecular Docking and Machine Learning: Datasets

The 3D structures of siramesine-related compounds were taken from the PubChem website. Among 88 compounds, 60 were picked up as similar to siramesine. Here, the similarity rule was based on (not) containing some structural features (see comment in Appendix A). These 60 compounds were labeled (S1–S60 in Appendix A with corresponding PubChem CID) and together with siramesine represent the siramesine 61 compound extrapolation set (S61-EX set) for in silico candidate selection.

The 5HK1 training and test set comprised 80 known ligands with known experimental pKi values from already published articles on the sigma 1 receptor QSAR [20] plus three additional literature ligands with known pKi values (PD144418, pentazocine and 4-IBP) taken from [37], making altogether an 83-compound sigma 1 receptor working set (Sig-W set). One-fifth of the dataset was taken as a test set and the rest was a training set.

The Erg2 ligand dataset contained 44 known literature ligands with known experimental pKi values taken from references [18,21]. All compounds with their corresponding pKi values are listed in the Appendix A and represent a 44-compound Erg-2 working set (Erg-W set). Because of the relatively small number of samples, the Erg-W set contains only the training set.

#### 3.2.3. Molecular Docking

Each compound (from each dataset) was downloaded from PubChem in 3D SDF format if available (if the SDF file was not available then its structure was built in the Avogadro program) and then the SDF file was copied and converted to PDB format. PDB format was used for molecular docking simulations in Autodock4 while SDF was utilized in the Gold program.

Concerning both 5HK1 and Erg2, only chain A was used for docking stripped of all ligands and water molecules. For 5HK1 grid options, the center coordinates were the arithmetic means of the coordinates of the experimentally crystallized ligand PD14418. Regarding Erg2 grid options, the center coordinates were the arithmetic means of all protein atoms and fall near the protein’s active site. For both proteins, cube size was 26 Å in Autodock4 (0.375 spacing with 70 × 70 × 70) and in the Gold all atoms were selected within 26 Å.

##### Sigma-1 Receptor Docking in Autodock4

For each ligand, rigid docking with Autodock4 defaults was conducted except that instead of only 10 GA runs, 100 GA runs were used. The aim was to get as many conformations as possible with the hydrogen bond (H-bond) between the positively charged tertiary NH^+^ of the ligand (nitrogen at pH 7.4 in dimethylamine, morpholine or piperidine is protonated) and any of the two possible negative carboxylate oxygen atoms of Glu-172 side chain residue. The criterion was as many as possible conformations with d(O...H) < 1.8 Å and d(O...H-N) < 2.9 Å and then we selected the lowest energy conformation among all such H-bond conformations. If there was no such conformation, then we selected the conformation with a hydrogen bond as shorter as possible with the upper limit of d(O...H) < 2.5 Å and d(O...H-N) < 3.5. If there was no conformation even with such an upper limit, then we repeated the whole docking for that ligand but with flexible docking instead of rigid docking. There were nine flexible docking side chains selected for flexible 5HK1 docking based on the highest interaction occurrence used in [20]: Phe-83, Leu-96, Tyr-103, Phe-107, Tyr-120, Glu-172, Thr-181, Leu-182 and Tyr-206. All Sig-W set compounds and all S61-EX set compounds got at least one H-bond conformation. When such (lowest energy) conformation was selected for each ligand, all the structures were converted from DLG file format to PDB and then finally (using Open Babel) to 3D SDF format suitable for PADEL descriptor generation.

##### Sigma-1 Receptor Docking in Gold

The Chemscore kinase template was loaded and used for the scoring function, without rescore, and without early termination. Flexible docking was used for all analyses with the following 10 flexible amino acid residues: Trp-89, Met-93, Tyr-103, Phe-107, Tyr-120, Trp-164, Glu-172, Thr-181, Leu-182 and Tyr-206. For each ligand of Sig-W set, 50 GA runs were used while for the S61-EX set, 200 GA runs were used, with default (i.e., automatic) GA settings. The same selection criterion for optimal 3D conformation used for PADEL descriptor generation was applied as in the former section with an addition that if 50 GA runs were not enough, then 200 GA runs were used. Only one sample of the S61-EX set did not produce any H-bond conformation with the upper limit of d(O...H) < 2.5 Å and d(O...H-N) < 3.5 and was discarded from further consideration; all other samples of both datasets got at least one H-bond conformation. Since the output result was already in 3D SDF format, it was directly utilized in descriptor generation.

##### Parallel but Independent Molecular Docking Analyses

All analyses (for both proteins) using Autodock4 were independent from analyses using Gold. Only the final results regarding candidate selection (see the last Section 2) were compared between these two programs. So, molecular descriptor datasets (Sig-W set and S61-EX) obtained by 3D Autodock4 were treated independently from the corresponding datasets obtained by Gold docking.

##### Sigma-1 Receptor QSAR

For all structures, all Padel 1D, 2D and 3D descriptors were selected with retained 3D coordinates; maximum running time per molecule was 10 s. Obtained 1875 molecular descriptors were preprocessed in a way that the Sig-W set and S61-EX were merged and each descriptor with a missing value for any compound among these two datasets was deleted. In the second step, all descriptors with less than three unique values were deleted. The third step in variable preprocessing defines the applicability domain to make Sig-W set models applicable to the S61-EX set, where each descriptor for each compound within the S61-EX set must have a value within the range of minimum and maximum value of the Sig-W set for the descriptor to be kept and considered further, otherwise if any compound in the S61-EX set contains a value outside the min-max range of the Sig-W set, that descriptor is discarded from further analysis. Finally, the data was mean-centered and scaled according to the Sig-W set.

Using such preprocessed descriptors in the former paragraph as independent variables and experimental pKi values as dependent variables, uninformative-variable elimination (UVE) partial-least square regression (UVE-PLS) was carried out to build the model on the Sig-W training set. Uninformative variable elimination (UVE) was used to filter noisy variables to optimize the prediction error. That was accomplished using the stability values for each variable, defined as the ratio of the mean regression coefficient of the corresponding variable and its standard deviation in the cross-validation procedure. This was because during the cross-validation procedure, the regression coefficient for each variable varied as different sample was left out [38]. At the end of cross-validation, a mean and standard deviation were obtained for each descriptor, and the ratio of the mean and standard deviation defines the descriptor’s stability. Having determined the stability values for all variables, the stability cutoff was set to remove all the variables whose stability was lower than such cutoff stability. The higher the cutoff is set, the more variables are excluded. For that reason, the cutoff level could be optimized. In building the model of the Sig-W set for both docking programs, the optimal model attained less than 200 original variables. The error estimation was calculated using a root-mean-square error of cross-validation RMSECV, which was also used to select the optimal number of latent variables. Finally, when the model with an optimal cutoff and number of latent variables was selected and used to predict the training and test pKi values and to obtain root-mean-square error of calibration (RMSEC) and validation RMSEP (with corresponding correlation coefficients between experimental pKi and predicted pKi values), it was also used to predict the pKi of the S61-EX set.

With the predicted pKi of S61-EX set, the compounds with a pKi higher than siramesine (see Section 2) were determined to be the sigma-1 receptor candidate compounds. All such candidate compounds were considered for the next QSAR modeling with Erg2 protein, labeled as a set of siramesine candidates for Erg2 (S-Erg2 set); while the other compounds from the S61-EX set were disregarded from further consideration. The new docking and QSAR cycle follows in the sections below.

##### Erg2 Receptor Docking in Autodock4

For each ligand in the Erg-W set, flexible docking with Autodock4 using default parametrization was conducted on the Erg2 protein with 100 GA runs. Before flexible docking, rigid docking was carried out using 100 GA runs with the 10 highest pKi ligands of the Erg-W set to determine residues which were most frequently interacting within the top 20 energy conformations of these 10 ligands. Each interaction is defined as ligand-protein atom-to-atom distance < 4 Å. Figure 4 displays fractions of all interactions. The most interacting Erg2 amino acids were: Asn-86, Tyr-105 (with the highest number of interactions), Ile-107, Thr-119, His-122, Leu-156, Tyr-164, Glu-174, Met-183 and Leu-184. These ten amino acids were used in flexible docking for all those ligands whose overall maximum number of torsions (including these 10 residues) was lower than 32. In case the number of torsion angles exceeded 32, then instead of ten, the following four flexible residues were used in molecular docking: Tyr-105, Tyr-164, His-122 (which are the top three interacting residues) and Glu-174 (key interacting residue). For the 32 Erg-W set compounds that had positive nitrogen at pH 7.4, the same H-bond-minimum energy conformation criterion was applied as in former sections (with sigma-1 receptor docking). If 100 GA runs with flexible docking did not form H-bonds, then rigid docking was also considered. For the rest of the 12 compounds without a positive nitrogen group at pH 7.4, simply the minimum energy conformation was selected (among 100 conformations). The S-Erg2 set contained 23 compounds (siramesine and 22 siramesine-related compounds which passed the prior QSAR sigma-1 receptor cycle) and underwent the same flexible docking protocol. All 32 Erg-W compounds and all 23 S-Erg2 set compounds attained at least one conformation with the H-bond with Glu-174 side chain residue. With the lowest energy conformation selected for each ligand, all such structures were converted from DLG files to 3D SDF format suitable for PADEL descriptor generation.

##### Erg2 Receptor Docking in Gold

The Chemscore kinase template was loaded and used as a scoring function, without rescore, and without early termination. Flexible docking was used for all analyses with the already mentioned 10 flexible amino acid residues in the former section. For each ligand of both the Sig-W set and S-Erg2 set, 200 GA runs were used. Please note that this S-Erg2 set was slightly different for Gold since selected compounds for the sigma-1 receptor for Gold differed slightly from those selected with Autodock4. The same selection criteria for optimal 3D conformation used for PADEL descriptor generation were applied as in the former section regarding 32 compounds with positive nitrogen and 12 compounds without such nitrogen. All 32 Erg-W compounds and all 23 S-Erg2 set compounds attained at least one conformation with an H-bond with Glu-174 side chain residue. The output result was already in 3D SDF format and it was directly used in descriptor generation.

##### Erg2 QSAR

The same protocol as for the Sigma-1 receptor was carried with only the following differences. The first was (as already stated), since there were only 44 compounds, Erg-W only contained the training set so there were no RMSEP and R^2^(val) values, instead leave-one-out cross-validation was carried out. RMSECV (instead of RMSEP) was used as a standard error of estimated pKi-s. The second difference was that the optimal model attained more than 200 variables, and due to having too many variables that model had to be neglected; instead another model with (slightly) higher error but with less than 200 variables was used, which luckily did not influence RMSECV values too much.

## 4. Conclusions

There are three classes of antifungal therapeutics available in clinics for yeast infection treatment (azoles, polyenes and echinocandins) and the need for novel treatment options due to emerging resistance of fungal species is growing. The high costs and research and development durability of new drug development have raised awareness of drug repositioning, reprofiling and retasking. Here we show strong in vitro activity of siramesine, both on planktonic cells and biofilm formed of clinically relevant Candida species. We hypothesize that antifungal activity of siramesine is directly connected with modulation of ergosterol biosynthesis. Based on the results obtained, the Erg-2 protein is highly suspected to be siramesine’s target, but further analyses on its potential site/mechanism(s) of action are needed. However, our results on its strong anti-Candida activity, and given its unsatisfactory anxiolytic properties but favourable innoxious properties, we propose further studies on eventual repurposing.

Besides experimental study related to siramesine in vitro activity, we provide also systematic docking and QSAR modeling of siramesine binding to Erg-2 as its most probable target. An additional result of this in silico study is a set of compounds which might be even stronger inhibitors of Erg-2, and may serve for experimental prioritization in future research on siramesine-related antifungal activity.

Since siramesine was already in Phase ll clinical trials [39] and is proven to be nontoxic and well tolerated in humans, our novel findings suggest its alternative use as an antifungal drug, although further analyses of its action are needed to elucidate and prove its efficacy and molecular targets.

## Figures and Tables

**Figure 1 molecules-26-03504-f001:**
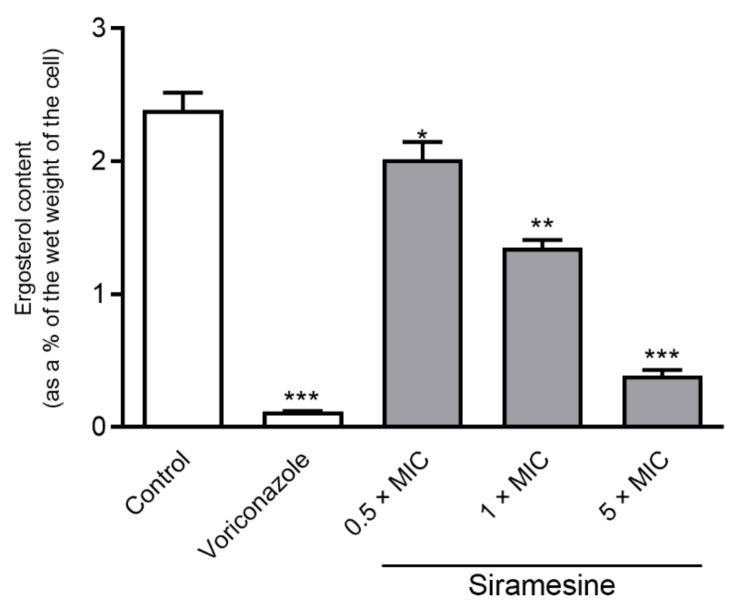
Modulation of ergosterol biosynthesis by siramesine on *C. albicans* ATCC 90028 (* *p* > 0.05; ** *p* < 0.05; *** *p* < 0.001 in comparison to control).

**Figure 2 molecules-26-03504-f002:**
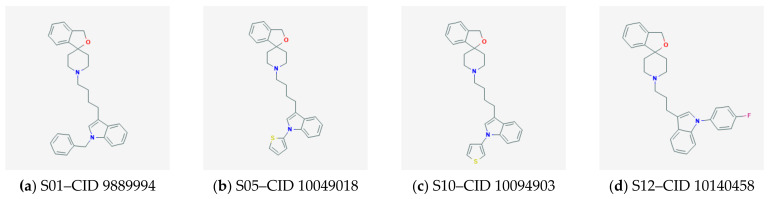
Selected 7 candidate compounds with corresponding CIDs and siramesine.

**Figure 3 molecules-26-03504-f003:**
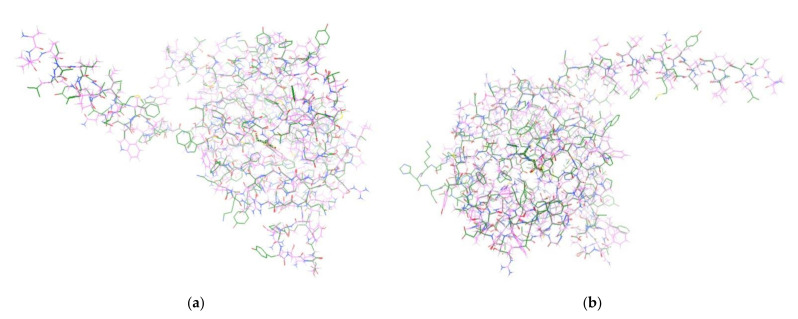
Alignment of sigma-1 receptor (pink) and Erg2 protein (green), viewed from two different angles (**a**,**b**).

**Figure 4 molecules-26-03504-f004:**
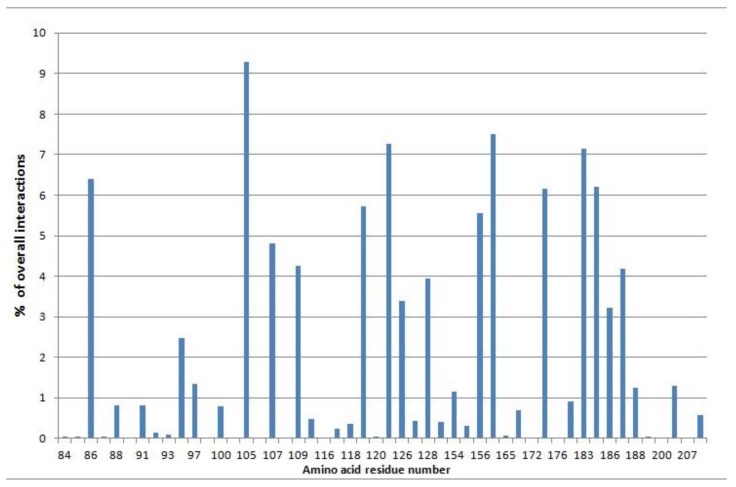
Interaction spectrum of considered top 200 conformations (20 conformation × 10 ten top pKi Erg-W set compounds) with rigid docking in Autodock4 program. Each interacting amino acid residue of Erg2 target with its interaction fraction (only interactions ligand-protein < 4Å are counted). Total number of interactions = 117,984.

**Table 1 molecules-26-03504-t001:** Antifungal capability of siramesine and siramesine-derived compounds against planktonic and biofilm-formed cells.

Fungal Species	Talarozole		Ozagrel		Siramesine		L-778123		MBX2982	
	MIC *	MBEC **	MIC	MBEC	MIC	MBEC	MIC	MBEC	MIC	MBEC
*Candida albicans* ATCC 10231	250	500	250	500	12.5	50	250	500	250	500
*Candida albicans* ATCC 90028	250	500	250	500	12.5	50	250	500	250	500
*Candida tropicalis* ATCC 750	250	500	250	500	12.5	62.5	250	500	>250	>500
*Candida kefyr* ATCC 2512	250	500	250	500	12.5	62.5	250	500	>250	>500
*Candida parapsylosis* ATCC 22019	250	500	250	500	12.5	50	250	500	>250	>500
*Candida krusei* ATCC 14243	250	500	250	500	12.5	50	250	500	250	500
*Issatchenkia orientalis* ATCC 6258	>250	>500	>250	>500	>250	>500	>250	>500	>250	>500
*Aspergillus brasiliensis* ATCC 16404	>250	NP ***	>250	NP	>250	NP	>250	NP	>250	NP

Legend: * MIC–minimal inhibitory concentration of planktonic fungal cells; ** MBEC–minimal biofilm-eradication concentration; *** NP–not performed; all data expressed as means on three independent measurements.

**Table 2 molecules-26-03504-t002:** Validated prediction of pKi values on the test set of σ1-receptor.

Comp	Exp pKi	Gold pKi	Adt pKi
**3**	7.41	7.39	7.59
**8**	6	5.83	6.24
**13**	7.72	7.15	7.29
**18**	8.1	8.38	8.31
**23**	8.27	8.91	8.76
**28**	9.07	8.24	8.32
**33**	8.89	8.67	8.68
**38**	8.11	7.91	7.93
**43**	6	6.63	6.30
**48**	6	6.58	6.74
**53**	7.2	6.78	7.35
**58**	8.16	8.65	8.89
**63**	7.12	7.25	7.01
**68**	6.37	6.75	6.89
**73**	8	7.89	7.89
**78**	7.21	7.18	7.47
Pentazocine	7.66	8.18	8.36

**Table 3 molecules-26-03504-t003:** Performance of established QSAR models using UVE-PLS.

Data	pKi Range	No. var	LV	RMSEC	*R*^2^ (Train)	RMSECV	RMSEP	*R*^2^ (val/CVtr)
σ1 Gold	6.0–9.16	99	4	0.401	0.821	0.514	0.436	0.788 (val)
σ1 Adt	6.0–9.16	140	4	0.397	0.824	0.526	0.436	0.813 (val)
Erg2 Gold	4.0–10.3	191	8	0.380	0.961	0.832	- *	0.774 * (CVtr)
Erg2 Adt	4.0–10.3	162	9	0.284	0.979	0.697	- *	0.841 * (CVtr)

* The Erg2 dataset only contained training samples, so instead of RMSEP and *R*^2^(val), the prediction error was estimated from leave-one-out cross-validation on the training set, i.e., RMSECV and *R*^2^ (CVtr).

**Table 4 molecules-26-03504-t004:** Predicted pKi of S61-EX dataset.

Comp	Gold pKi	Adt pKi	Comp	Gold pKi	Adt pKi	Comp	Gold pKi	Adt pKi
Siramesine	7.24	7.22	S21	7.40	7.23	S45 * (G&A)	8.48	8.21
S01 * (G&A)	8.19	8.21	S22 * (A)	7.51	7.72	S46	7.46	7.26
S02 * (G&A)	8.28	8.14	S23	6.68	7.06	S47 * (G&A)	7.73	7.83
S03	6.32	6.89	S24 * (G&A)	7.88	7.90	S48 * (G&A)	7.70	7.99
S04	6.95	7.05	S25 * (G&A)	7.80	7.75	S49	7.14	7.32
S05 * (G&A)	8.08	7.88	S26	7.08	7.15	S54	6.97	7.35
S06	7.25	7.44	S27	7.07	7.17	S55	7.25	7.24
S07	7.34	7.66	S28 * (G&A)	7.76	7.87	S56	7.19	7.26
S08 * (G&A)	8.22	8.14	S29 * (G&A)	8.25	7.85	S57	7.14	7.27
S09	7.31	7.40	S30 * (G&A)	7.81	7.93	S58	7.04	6.98
S10 * (G&A)	8.20	8.45	S31	6.50	7.30	S59	6.77	6.88
S11	7.27	7.19	S33	7.15	7.39	S62	6.62	7.22
S12 * (G&A)	8.05	8.08	S34	-	7.32	S63	6.61	7.28
S13	7.08	7.41	S36 * (G)	7.70	7.57	S64	6.65	6.64
S14 * (G&A)	7.77	8.06	S37	7.15	7.22	S65	7.48	7.46
S15	7.33	7.52	S38 * (G&A)	7.71	7.75	S71	7.67	7.19
S16	6.82	7.20	S39 * (G&A)	8.17	7.95	S72	7.66	7.42
S17 * (G)	7.77	7.62	S40	7.01	7.17	S80 * (G&A)	8.58	8.59
S18	7.35	7.22	S41 * (G&A)	8.17	8.17	S84 * (A)	7.19	7.78
S19	6.81	6.80	S43	7.54	7.54			
S20 * (G&A)	8.49	8.63	S44	7.05	7.00	Stand. err.	0.44	0.44

* Denotes significantly higher pKi than siramesine’s pKi, i.e., higher pKi than the sum of siramesine’s pKi and standard error (e.g., pKi(S02, Gold, 8.28) > 7.24 + 0.44), number in brackets (‘G&A’ or ‘A’) represents whether both programs obtained significantly higher pKi.

**Table 5 molecules-26-03504-t005:** Cross-validated prediction of pKi ligands of Erg2 target.

Compound	Exp pKi	Gold pKi	Adt pKi	Compound	Exp pKi	Gold pKi	Adt pKi
1,3-di-o-tolylguanidine	5.7	6.82	5.73	MDL-28815	9.4	8.68	9.37
carbisocaine	5.5	6.84	5.79	Pentazocine	6	5.33	6.36
AY-9944	7.19	7.84	7.89	jervine	6.55	5.94	5.90
buflumedil	5.15	2.85	3.67	progesterone	5.35	5.29	5.62
BM-15766	4.78	6.38	6.97	opipramol	7.77	6.22	6.12
azidopamil *	5 *	5.73	4.99	SR-31747	8.05	8.54	8.20
amiodarone	7.21	6.85	6.75	ronipamil	7.89	7.84	7.48
3-PPP	5.92	6.20	6.43	terbinafine *	4.3 *	5.95	5.17
enclomiphene	6.8	7.76	8.13	tamoxifene	5.82	6.20	5.85
CP-74932-4	4.5	4.89	3.61	raloxifene	7.18	7.23	7.38
Haloperidol	9.3	9.25	8.96	solanidine *	4 *	3.78	3.62
Emopamil	7.19	7.52	7.14	solasodine	5.93	6.05	5.87
fenpropimorph	10.3	9.71	9.14	testosterone	5.11	5.18	4.42
L-690404	8.3	8.72	8.03	triparanol	8.7	8.47	8.31
ifenprodil	9	7.73	8.48	testosterone propionate	5.72	5.23	5.69
N-Allylnormetazocine	5.33	5.62	5.45	tridemorph	10.05	10.00	10.39
cyclopamine	6.3	6.95	7.12	trifluoperazine	6.3	5.85	6.00
naftifine	6.51	4.72	5.65	trifluperidol	9.82	10.04	9.62
corticosterone *	4 *	4.97	5.21	U-18666A	9.7	9.23	10.20
R-59494	5.55	5.02	5.71	VUF-8410	6.77	7.38	7.17
MDL-5332	9.15	8.72	8.66	zuclomiphene	8.7	7.69	8.57
nafoxidine	6.63	6.95	6.54	tomatidine	6.41	7.30	6.50

* Marked “exp values” of pKi in literature for few compounds are designated to be less than rather than equal to mentioned. For example, in [21] Ki(solanidine) > 10^–4^ M (i.e., pKi < 4). Here values for corticosterone, azidopamil, terbinafine, and solanidine are taken to be equal to mentioned values in literature [18,21].

**Table 6 molecules-26-03504-t006:** Predicted pKi values of S-Erg2 dataset for both Adt and Gold.

Comp	Gold pKi	Adt pKi	Comp	Gold pKi	Adt pKi	Comp	Gold pKi	Adt pKi
Siramesine	6.63	7.13	S20	7.47 *	7.64	S39	5.89	6.51
S01 **	7.97 *	8.43 *	S22	-	7.75	S41 **	7.92 *	8.21 *
S02	7.15	7.10	S24	7.32	7.02	S45	6.75	6.79
S05 **	7.59 *	7.96 *	S25	7.57*	7.77	S47	7.21	5.73
S08	7.36	6.98	S28	5.97	7.04	S48	6.09	6.42
S10 **	7.55 *	9.07 *	S29 **	7.61 *	7.93 *	S80	7.80 *	7.83
S12 **	7.69 *	8.05 *	S30 **	7.51 *	8.11 *	S84	-	6.72
S14	7.06	8.32 *	S36	7.21	-			
S17	6.29	-	S38	6.63	6.91	Stand. err.	0.83	0.70

* Denotes significantly higher pKi than siramesine’s pKi, i.e., higher pKi than the sum of siramesine’s pKi and standard error (e.g., pKi(S01, Adt, 8.43) > 7.13 + 0.70). ** Siramesine-related compounds with significantly different pKi values predicted from both programs, and also that passed both selection cycles. Therefore, they are suggested as candidate compounds.

## Data Availability

The data presented in this study are available in Appendix A.

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
