# Peer review of "In Vitro Confirmation of Siramesine as a Novel Antifungal Agent with In Silico Lead Proposals of Structurally Related Antifungals"

_molecules, 2021, doi:10.3390/molecules26123504_

Round 1
Reviewer 1 Report
Title: Siramesine and Siramesine-related Compounds as Novel Antifungal Agents
Title: personally I think the title need to reflect better the conducted study. The current title is too general.
Pls be specific , is it as antifungal or anti-yeast ? Yeast and fungi are two types of organisms, which belong to the kingdom Fungi. Yeast is a type of fungi, which is a unicellular, oval-shaped organism. Fungi are mostly multicellular, consisting of fungal hyphae.
Abstract:
Limited number of medicinal products available for the treatment of fungal infections makes control of fungal pathogens problematic, especially since number of fungal resistance incidents increases. Given that high costs and slow development of new antifungal treatment options, repurposing of already known compounds is one of proposed strategies. Our study identifies siramesine and several siramesine-related compounds as potential antifungals. This novel indication was suggested in our previous in-silico study and confirmed through in vitro testing using several yeasts species and mold. The results showed susceptibility of Candida species to siramesine with MIC at concentration 12.5 µg/mL, whereas other candidates had no antifungal activity. Siramesine was also effective against in vitro biofilm formation and already formed biofilm was reduced following 24-h treatment with MBIC range from 50-62.5 µg/mL. Siramesine is involved in modulation of ergosterol biosynthesis in vitro, which indicates possible target for its antifungal activity. This implicates the possibility of siramesine repurposing especially since already published data about toxicity. Furthermore, we provide in-depth in-silico analysis of the siramesine and siramesine similar compounds, providing extended lead set for further preclinical and clinical investigation, needed to clearly define molecular targets and to elucidate its in vivo effectiveness as well.
Comments: Pls include objective of the study in the abstract. Pls justify the term repurposing in this context.
The whole manuscript is very well written with complete information related to experiment design, implementation and the corresponding findings.
Nevertheless, I have some minor comments:
Introduction
Current treatment options for fungal diseases are unsatisfactory-this statement is inappropriate, pls explain better
Conclusion:
Since siramesine was already in Phase ll clinical trials [5] and is proved to be non-toxic and well tolerated in humans, our novel findings suggest its alternative use in clinics-this statement is not substantiated, pls remove or rephrase.
Author Response
Response to reviewer 1 comments
Reviewer 1.
Title: personally I think the title need to reflect better the conducted study. The current title is too general.
Response: We kindly thank for the suggestion. We revised the title accordingly.
Pls be specific , is it as antifungal or anti-yeast ? Yeast and fungi are two types of organisms, which belong to the kingdom Fungi. Yeast is a type of fungi, which is a unicellular, oval-shaped organism. Fungi are mostly multicellular, consisting of fungal hyphae.
Response: The reviewer is of course correct as unicelular fungi are yeasts. However, since anti-yeast drugs is an unusual term in literature, we opted to use the term antifungal as this is usual term in literature, also in cases when actual activity of the drugs/compounds is assessed against yeasts.
However, throughout the paper when addressing the model organism we used the proper term – yeast.
Thus, the paper has been revised accordingly.
Abstract:
Limited number of medicinal products available for the treatment of fungal infections makes control of fungal pathogens problematic, especially since number of fungal resistance incidents increases. Given that high costs and slow development of new antifungal treatment options, repurposing of already known compounds is one of proposed strategies. Our study identifies siramesine and several siramesine-related compounds as potential antifungals. This novel indication was suggested in our previous in-silico study and confirmed through in vitro testing using several yeasts species and mold. The results showed susceptibility of Candida species to siramesine with MIC at concentration 12.5 µg/mL, whereas other candidates had no antifungal activity. Siramesine was also effective against in vitro biofilm formation and already formed biofilm was reduced following 24-h treatment with MBIC range from 50-62.5 µg/mL. Siramesine is involved in modulation of ergosterol biosynthesis in vitro, which indicates possible target for its antifungal activity. This implicates the possibility of siramesine repurposing especially since already published data about toxicity. Furthermore, we provide in-depth in-silico analysis of the siramesine and siramesine similar compounds, providing extended lead set for further preclinical and clinical investigation, needed to clearly define molecular targets and to elucidate its in vivo effectiveness as well.
Comments: Pls include objective of the study in the abstract. Pls justify the term repurposing in this context.
Response: We included the aim of our study into Abstract and clarified the term repurposing in the Introduction.
The whole manuscript is very well written with complete information related to experiment design, implementation and the corresponding findings.
Nevertheless, I have some minor comments:
Introduction
Current treatment options for fungal diseases are unsatisfactory-this statement is inappropriate, pls explain better
Response: We introduced new paragraph in Introduction to better explain aim and background of our study.
Conclusion:
Since siramesine was already in Phase ll clinical trials [5] and is proved to be non-toxic and well tolerated in humans, our novel findings suggest its alternative use in clinics-this statement is not substantiated, pls remove or rephrase.
Response: We rephrased this statement in accordance with the suggestion.
We kindly thank rewiever for time spent to make our work better and appropriate to general scientific community.
Reviewer 2 Report
Dear authors
The limited number of medicines available to treat fungal infections makes controlling fungal pathogens problematic, especially as the number of fungal resistance incidents increases. Given the high costs and slow development of new antifungal treatment options, the reuse of already known compounds is one of the proposed strategies. The study proposed by the authors identifies siramesine and several compounds related to siramesine as potential antifungals. This new indication was suggested in our previous in silico study and confirmed through in vitro tests using different species of yeasts and molds. The results showed the susceptibility of Candida species to siramesine with MIC at the concentration of 12.5 µg / mL, while other candidates had no antifungal activity. Siramesine was also effective against biofilm formation in vitro and the already formed biofilm was reduced after 24-hour treatment with an MBIC range of 50 to 62.5 µg / mL. Siramesine is involved in the modulation of ergosterol biosynthesis in vitro, which indicates a possible target for its antifungal activity. This implies the possibility of reuse of siramesine, especially since the data already published on toxicity. Additionally, we provide in-depth in silico analysis of siramesine and siramesine-like compounds, providing an extended lead set for further preclinical and clinical investigations, necessary to clearly define molecular targets and also to elucidate its efficacy in vivo.
1 - Introduction: must be reformed in the content and in the writing of the general part review the syntax of the topic.
2- methods: The methods are clear and well applied to the topic of the paper.
3- Discussion : to deepen in consideration of the problem of antifungal resistance the use of essential oils against multidrug-resistant strains of Candida sp. and clinical applications. Learn more about this by using and citing the following references:
- DOI: https://doi.org/10.3390/jof7050383
- DOI: 10.3390/molecules25061463
- DOI: 10.1055/a-1201-3375
4 - Check the bibliographic entries throughout the text, some of which are non-compliant, review some entries in the references and necessarily insert those referred to in point 3 for the purpose of acceptance by me.
5 - Review the English grammar and in particular the applied scientific English: in particular, verbal tenses and syntax in the discussion.
Author Response
Response to Reviewer 2 comments
Rewiewer 2.
The limited number of medicines available to treat fungal infections makes controlling fungal pathogens problematic, especially as the number of fungal resistance incidents increases. Given the high costs and slow development of new antifungal treatment options, the reuse of already known compounds is one of the proposed strategies. The study proposed by the authors identifies siramesine and several compounds related to siramesine as potential antifungals. This new indication was suggested in our previous in silico study and confirmed through in vitro tests using different species of yeasts and molds. The results showed the susceptibility of Candida species to siramesine with MIC at the concentration of 12.5 µg / mL, while other candidates had no antifungal activity. Siramesine was also effective against biofilm formation in vitro and the already formed biofilm was reduced after 24-hour treatment with an MBIC range of 50 to 62.5 µg / mL. Siramesine is involved in the modulation of ergosterol biosynthesis in vitro, which indicates a possible target for its antifungal activity. This implies the possibility of reuse of siramesine, especially since the data already published on toxicity. Additionally, we provide in-depth in silico analysis of siramesine and siramesine-like compounds, providing an extended lead set for further preclinical and clinical investigations, necessary to clearly define molecular targets and also to elucidate its efficacy in vivo.
1 - Introduction: must be reformed in the content and in the writing of the general part review the syntax of the topic.
Response: We kindly thank for this comment as this improved manuscript in general. Some part of text have been added.
2- methods: The methods are clear and well applied to the topic of the paper.
3- Discussion : to deepen in consideration of the problem of antifungal resistance the use of essential oils against multidrug-resistant strains of Candida sp. and clinical applications. Learn more about this by using and citing the following references:
DOI: https://doi.org/10.3390/jof7050383
DOI: 10.3390/molecules25061463
DOI: 10.1055/a-1201-3375
Response: Search for novel drugs among herbal-related drugs and substances is indeed one of possibilities to overcome problem of antifungal resistance. We kindly thank reviewer to suggest this important research area. After careful reading of suggested papers we introduced text related to the use of essential olis against resistant candida strains.
4 - Check the bibliographic entries throughout the text, some of which are non-compliant, review some entries in the references and necessarily insert those referred to in point 3 for the purpose of acceptance by me.
Response: We carefully checked all the references and we believe all of them are now in place. Thank you for the notification and suggestions.
5 - Review the English grammar and in particular the applied scientific English: in particular, verbal tenses and syntax in the discussion.
Response: Complete text has been edited.
We kindly thank reviewer for all notifications and suggestions, as those improved text bringing more clarity and broadening scope for scientific community. We hope that our findings will be useful for the introduction of novel drugs or treatments in future.l.